# Coefficients of Thermal Expansion in La_3_Ga_5_SiO_14_ and Ca_3_TaGa_3_Si_2_O_14_ Crystals

**DOI:** 10.3390/ma16124470

**Published:** 2023-06-19

**Authors:** Dmitry Roshchupkin, Dmitry Kovalev

**Affiliations:** 1Institute of Microelectronics Technology and High Purity Materials, Russian Academy of Sciences, 142432 Chernogolovka, Russia; 2Merzhanov Institute of Structural Macrokinetics and Material Science, Russian Academy of Sciences, 142432 Chernogolovka, Russia; kovalev@ism.ac.ru

**Keywords:** langasite, thermal expansion coefficients, X-ray powder diffraction, piezoelectric crystals, Czochralski method

## Abstract

The ordered Ca_3_TaGa_3_Si_2_O_14_ and disordered La_3_Ga_5_SiO_14_ crystals of the lantangallium silicate family were grown via the Czochralski method. The independent coefficients of thermal expansion of crystals $\alpha_{c}$ and $\alpha_{a}$ were determined using X-ray powder diffraction based on the analysis of X-ray diffraction spectra measured in the temperature range of 25~1000 °C. It is shown that, in the temperature range of 25~800 °C, the thermal expansion coefficients are linear. At temperatures above 800 °C, there is a nonlinear character of the thermal expansion coefficients, associated with a decrease in the Ga content in the crystal lattice.

## 1. Introduction

Crystals of the lantangallium silicate family La_3_Ga_5_SiO_14_ (langasite, LGS) are promising piezoelectric materials for telecommunication systems, for information transmission, and for processing in a real-time mode. LGS crystals are grown via the Czochralski method from a melt [1,2,3,4,5,6,7,8,9] and are of the point symmetry group 32, like SiO_2_ piezoquartz crystals. Their piezoelectric properties occupy an intermediate position between SiO_2_ piezoquartz crystals and LiNbO_3_ ferroelectric crystals [10,11,12]. In quartz crystals, the phase transition from α to β phase is observed at 575 °C and this process is nonreversible. Therefore, piezoquartz crystals are not suitable for high-temperature surfaces and bulk acoustic wave sensors that can operate at temperatures as high as 600 °C. On the other hand, there are no phase transitions in LGS crystals up to the melting point ($T_{m}\sim 1450$ °C). Moreover, LGS crystals have low surface acoustic wave velocities, which makes them attractive for miniature acoustoelectronic devices.

Crystals of the LGS family grown via the Czochralski method have a high perfection of the crystal lattice, which allows a wide range of studies using X-ray radiation. Usually, electrical measuring techniques are employed to analyze the piezoelectric and acoustic properties of crystals. In [13], high-resolution triple-axis X-ray diffraction was used to evaluate the piezoelectric moduli in crystals from the LGS family. This technique utilizes the alternation of the crystal’s unit cell parameters when an external electric field is applied, known as the inverse piezoelectric effect. Consequently, the interplanar spacing undergoes changes leading to a corresponding shift in the angular position of the Bragg peak. This shift enables us to determine the piezoelectric moduli by examining the change in the angular position of the Bragg peak. However, it should be noted that this approach was initially proposed in [14,15,16] for measuring the piezoelectric moduli in a quartz crystal. These works [14,15,16] utilize a double-crystal X-ray diffraction setup, which does not let us eliminate the contribution of deformations such as torsion and bending in the crystal when an external electric field is applied. In contrast, the use of a crystal-analyzer in the optical scheme of the triple-axis X-ray diffraction enables us to eliminate the contribution of the substrate deformation and to consider only the change of interplanar spacing during the measurement process.

X-ray topography and diffraction Methods are optimal for studying the process of acoustic wave propagation in solids. X-ray topography techniques on synchrotron radiation sources allow real-time visualization of the process of surface acoustic waves (SAW) propagation using stroboscopic X-ray topography [17,18,19,20] or the Talbot effect [21]. These topography methods can be applied to visualize the interaction process with crystal lattice defects, to measure the power flow angles, and to obtain the wavelength of the SAW. The investigation of the SAW propagation process in solids via high-resolution X-ray diffractometry is based on the process of X-ray diffraction on a crystal lattice, which is sinusoidally modulated by the SAW [22,23]. The presence of such a diffraction grating leads to the appearance of diffraction satellites around the Bragg peak on the rocking curve. The angular divergence between these diffraction satellites corresponds to the SAW wavelength, while their number and intensity are defined by the SAW amplitude. It is also possible to investigate the attenuation of SAW along the direction of propagation and within the crystal’s depth.

In this work, the thermal expansion coefficients (α) of La_3_Ga_5_SiO_14_ and Ca_3_TaGa_3_Si_2_O_14_ (CTGS) crystals were measured. Typically, to determine the coefficients of thermal expansion, bars are cut from crystals and the elongation of the bars is examined during heating. In this research, we employed X-ray diffraction to derive the thermal expansion coefficients of a crystal under controlled heating conditions. By analyzing the X-ray diffraction spectra, we can accurately extract the angular displacement of the Bragg peaks due to changes in the interplanar spacing. The change in the interplanar spacing, resulting from temperature fluctuations, enable us to calculate the value of the thermal expansion coefficients. Previously, α was measured in the crystal Ca_3_TaGa_3_Si_2_O_14_ based on a study of the elongation of bars cut from the crystal as the temperature varies [24,25,26].

## 2. Piezoelectric La_3_Ga_5_SiO_14_ and Ca_3_TaGa_3_Si_2_O_14_ Crystals

La_3_Ga_5_SiO_14_ and Ca_3_TaGa_3_Si_2_O_14_ crystals were grown from the melt using the Czochralski method to measure the thermal expansion coefficients. Figure 1 shows the disordered La_3_Ga_5_SiO_14_ crystal (a) and ordered Ca_3_TaGa_3_Si_2_O_14_ crystal (b) grown via the Czochralski method.

Thermal expansion coefficients α of the crystals of langasite family must be taken into account in the process of the fabrication of high-temperature sensors and acoustoelectronic and acousto-optic devices. It should be noted that, in crystals of point group symmetry 32, the X and Y directions are equivalent in terms of thermal expansion. Thus, there are two independent coefficients of thermal expansion in crystals of point group symmetry 32: along direction c ([001]) $\alpha_{c}$ and along direction a ([110]) $\alpha_{a}$.

To measure α, the powder thermo diffraction method was employed while subjecting the sample to varying temperatures. The grown crystals were ground into a fine powder with an average grain size of ~100 nm. The thermal expansion coefficient α in the Y−cut of an LGS crystal was also measured by measuring the rocking curves of the crystal at different temperatures.

## 3. Experimental Set-Up

The studies were performed on an ARL X’TRA X-ray diffractometer. The ANTON PAAR HTK 2000 high-temperature chamber with a vacuum of 6 × 10^−5^ mbar was implemented for high-temperature investigations. An X-ray optical scheme of the diffractometer is shown in Figure 2. An X-ray tube with a Cu anode was used as an X-ray source. The β−lines were filtered with an Ni filter. Then, the X-ray radiation was collimated by the entrance slit with the size of 1 mm. Collimated X-ray radiation was diffracted on a powder of grown crystals placed on a W-heater in a high-temperature vacuum chamber. Diffracted X-ray radiation was recorded with a standard NaI scintillation detector with an input slit size of 1 mm. The W-heater provides the ability to change the temperature from room temperature to 2300 °C. The investigations were carried out in the temperature range from 25 °C to 1200 °C.

X-ray diffraction spectra were measured in a Θ–2Θ diffraction scheme.

The thermal expansion coefficient was also measured separately by measuring the rocking curves of the crystal at different temperatures. Figure 3 shows a photograph of the experimental set-up on the KMC2 optical beamline of the BESSY II synchrotron radiation source.

X-rays radiation with an energy E = 10 keV (λ = 1.24 Å) was selected with a Si(111) double-crystal monochromator. Further, X-ray radiation diffracts onto the studied crystal, located in a high-temperature Be vacuum chamber. The diffracted X-ray radiation was recorded with a standard NaCl scintillation detector. Measurement of the rocking curves makes it possible to determine the interplanar spacing in the crystal. Changes in the crystal temperature lead to changes in the interplanar distance and a corresponding change in the position of the Bragg peak on the curve. Thus, this approach also makes it possible to measure thermal expansion coefficients.

## 4. Experimental Results

Figure 4 shows X-ray diffraction spectra of CTGS crystal powder measured at different temperatures. The figure shows that an increase in temperature leads to a shift in the angular position of the diffraction peaks toward smaller angles, which corresponds to an increase in the values of interplanar spacing and a corresponding change in the parameters of the crystal unit cell. However, it should be noted that the linear behavior of the diffraction peaks is observed up to 800 °C. At this point, there is an abrupt alteration in the position of the diffraction peaks, which is associated with the peculiarities of the behavior of LGS crystals at high temperatures. At temperatures above 800 °C, gallium ions begin to leave the surface of the langasite crystal family, which leads to the formation of a porous structure.

Figure 5 shows a micrograph of the Ca_3_TaGa_3_Si_2_O_14_ crystal surface obtained via scanning electron microscopy in the secondary electron emission mode before (a) and after (b) thermal treatment of the substrate at 1000 °C. In this case, a substrate was a Z−cut from the CTGS crystal, and its surface was polished so that the surface roughness did not exceed 3 Å. The substrate was subjected to heat treatment in a vacuum, but only at temperatures above 800 °C was there a significant transformation of the crystal surface: the surface became porous. In micrographs, one can observe significant surface degradation due to the escape of Ga ions from the crystal lattice to the surface, followed by evaporation at high temperatures. That is, in this case, there is a decrease in the number of Ga ions in the crystal lattice. Therefore, it is advisable to determine the coefficient of thermal expansion within the temperature range up to 800 °C, where there is a linear dependence of changes in the angular position of diffraction peaks on X-ray diffraction spectra.

Figure 6 shows the dependences of the crystal unit cell parameters a and c of the CTGS crystal in the temperature range from 25 °C to 1000 °C. The black circles correspond to experimental data based on the diffraction spectra in Figure 3. The red lines correspond to a linear approximation of the dependence of the alteration in the crystal unit cell parameters as a function of temperature. The two independent coefficients of thermal expansion, in our case, are defined as
(1)$$\alpha =\frac{∆d}{d}\times \frac{1}{∆T},$$
where d is the interplanar spacing at room temperature and d is the change in the interplanar spacing with the temperature change T. As noted earlier, crystals of the point symmetry group 32 have two independent coefficients of thermal expansion along direction c (direction [001]) $\alpha_{c}$ and direction a (direction [110]) $\alpha_{a}$. Thermal expansion coefficients of the CTGS crystal were determined based on the results of linear approximation of the change in the parameters of the crystal unit cell in the temperature range 25–800 °C. The values of the thermal expansion coefficients for the CTGS crystal calculated from expression (1) were $\alpha_{c}=6.58\times 10^{-6}$ K^−1^ and $\alpha_{a}=6.98\times 10^{-6}$ K^−1^, respectively. These values are in a good agreement with the values ($\alpha_{c}=6.8\times 10^{-6}$ K^−1^ and $\alpha_{a}=7.1\times 10^{-6}$ K^−1^) obtained in [24] when the change in the length of the CTGS crystal bar was measured at varied temperature conditions. It should also be noted that the article [26] demonstrated the nonlinear behavior of the coefficient of thermal expansion at temperatures below 0 °C. Additionally, present studies demonstrate a deviation of the coefficient of thermal expansion from linear at temperatures above 800 °C due to the decrease in Ga in the crystal lattice.

Figure 7 shows the diffraction spectra of the LGS crystal powder measured at 25 °C, 400 °C, 600 °C, 800 °C, 1000 °C, and 1200 °C. It can also be observed that the positions of the diffraction peaks shift with increasing temperature due to increasing interplanar spacing. In the temperature range of 25~800 °C, the change in the position of the diffraction peaks is linear, and at temperatures of 1000 °C and 1200 °C, there is a stronger change in the interplanar spacing due to the removal of Ga from the crystal structure.

Figure 8 shows the dependences of changes in the parameters of the crystal unit cell a (a) and c (b) of the LGS crystal in the temperature range 25~1200 °C. In the figure, the black circles correspond to the experimental results based on the obtained diffraction spectra of Figure 6. The red lines are a linear approximation of the change in the crystal unit cell parameters as a temperature function, which were used to measure the thermal expansion coefficients. The values of thermal expansion coefficients for the LGS crystal were $\alpha_{c}=6.08\times 10^{-6}$ K^−1^ and $\alpha_{a}=5.64\times 10^{-6}$ K^−1^.

Figure 9 shows the results of the study of X-ray diffraction on the Y−cut of an LGS crystal. In the Y−cut, the planes (100) are parallel to the crystal surface. The full width at huff maxima of the rocking curve is FWHM=0.0026°. Figure 9a shows the results of measuring the rocking curves at temperatures of 20 °C, 100 °C, 200 °C, 300 °C, and 400 °C. Increasing the temperature leads to a decrease in the Bragg angle value by increasing the interplanar spacing. Figure 9b shows the dependence of the change in the interplanar spacing versus temperature. Based on this graph, the coefficient of thermal expansion was determined, which was $\alpha_{a}=5.638\times 10^{-6}$ K^−1^. This approach is more accurate because it is based on the measurement of rocking curves, the half-width of which is significantly less in relation to the angular width of the peaks on the XRD spectra.

## 5. Conclusions

The thermal expansion coefficients were measured in crystals of the lantangallium silicate family (La_3_Ga_5_SiO_14_ and Ca_3_TaGa_3_Si_2_O_14_ crystals) via X-ray powder diffraction (CTGS: $\alpha_{c}=6.58\times 10^{-6}$ K^−1^ and $\alpha_{a}=6.98\times 10^{-6}$ K^−1^; LGS: $\alpha_{c}=6.08\times 10^{-6}$ K^−1^ and $\alpha_{a}=5.64\times 10^{-6}$ K^−1^) in a temperature range from 25 °C to 1200 °C. It is shown that the thermal expansion coefficients are linear in the temperature range 25~800 °C. At temperature above 800 °C, a non-linear behavior of the thermal expansion coefficients (a sharp increase in the thermal expansion coefficient) is observed, associated with a decrease in the Ga content in the crystal lattice of crystals of the lanthanum gallium silicate family.

The possibility of measuring the coefficient of thermal expansion in crystals based on the measurement of rocking curves under the conditions of temperature change has also been demonstrated. The method has a higher accuracy of measuring interplanar spacing compared to the XRD method.

## Figures and Tables

**Figure 1 materials-16-04470-f001:**
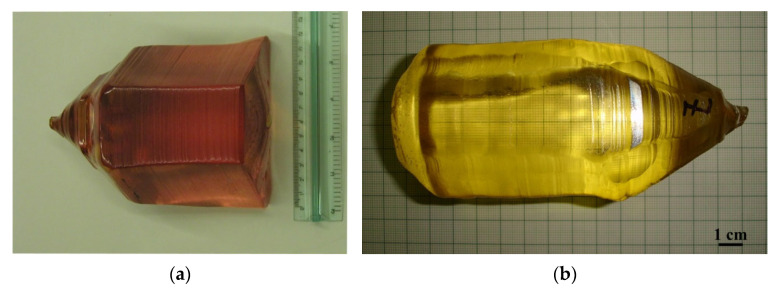
Crystals of the langasite family grown along axis c [001]: (**a**) 4″ disordered La_3_Ga_5_SiO_14_ crystal (LGS); (**b**) 3″ ordered Ca_3_TaGa_3_Si_2_O_14_ crystal (CTGS).

**Figure 2 materials-16-04470-f002:**
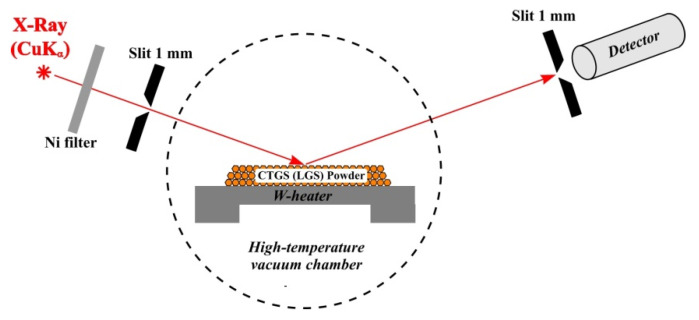
Experimental set-up of X-ray diffractometer ARL X’TRA.

**Figure 3 materials-16-04470-f003:**
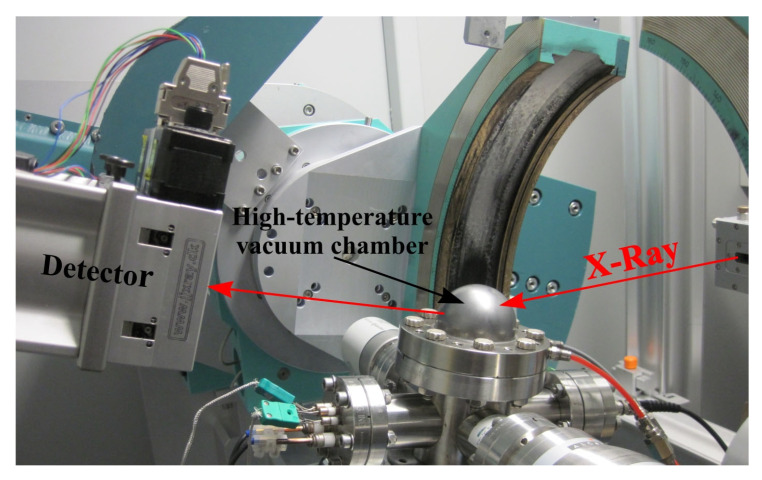
Experimental set-up of the double axis X-ray diffractometer on the KMC2 optical beamline of the BESSY II synchrotron radiation source.

**Figure 4 materials-16-04470-f004:**
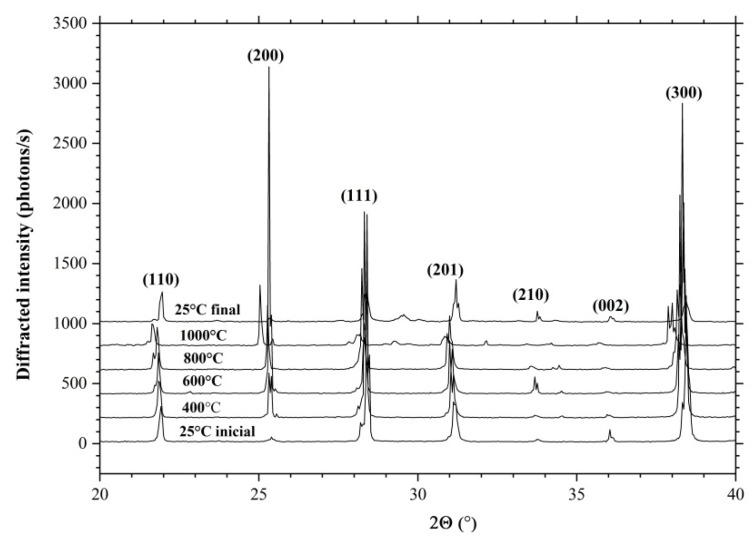
XRD spectra of CTGS crystal measured at 25 °C, 400 °C, 600 °C, 800 °C, and 1000 °C.

**Figure 5 materials-16-04470-f005:**
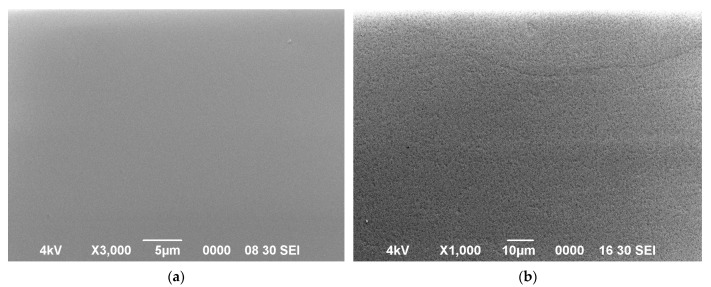
SEM image of LGS crystal surface before (**a**) and after heat treatment at 850 °C within 30 min (**b**).

**Figure 6 materials-16-04470-f006:**
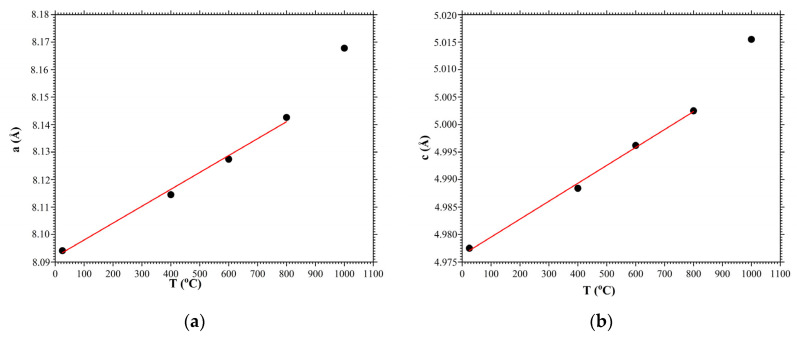
Dependences of the crystal unit cell parameters of CTGS crystal on temperature: (**a**) *a*, (**b**) *c*. Black circles are experimental values, red lines are linear approximation.

**Figure 7 materials-16-04470-f007:**
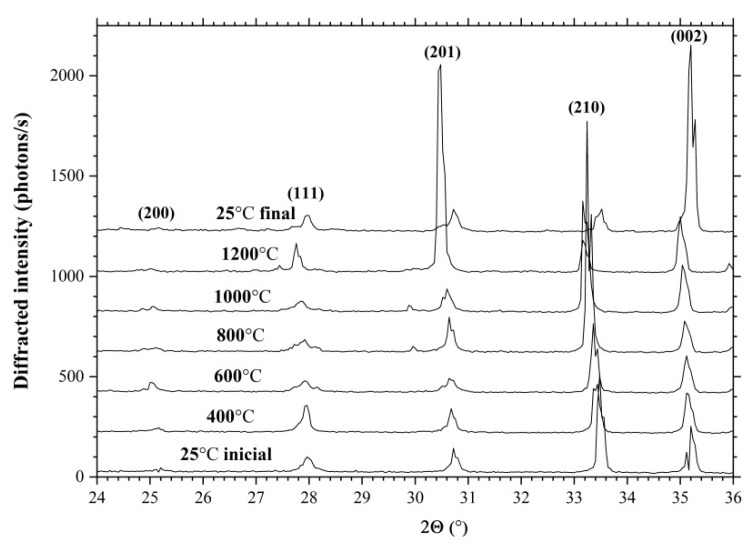
XRD spectra of LGS crystal measured at 25 °C, 400 °C, 600 °C, 800 °C, 1000 °C and 1200 °C.

**Figure 8 materials-16-04470-f008:**
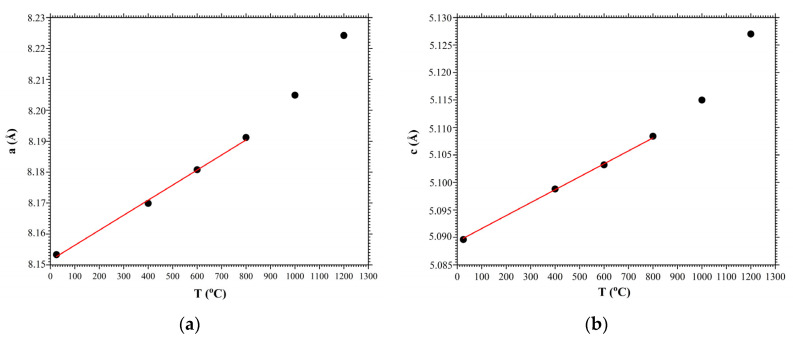
Dependences of the crystal unit cell parameters of LGS crystal on temperature: (**a**) *a*, (**b**) *c*. Black circles are experimental values, red lines are linear approximation.

**Figure 9 materials-16-04470-f009:**
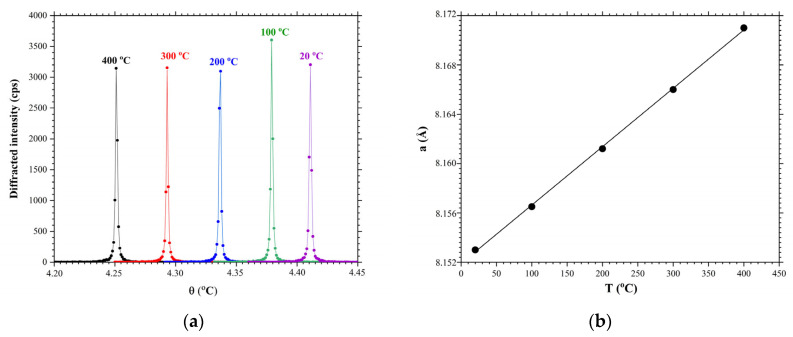
Rocking curves of the Y−cut of an LGS crystal measured at different temperature: (**a**) change of interplanar spacing versus temperature, (**b**) reflection from planes (100). Black circles are experimental values, black lines are linear approximation.

## Data Availability

All relevant data presented in the article are stored according to institutional requirements and as such are not available online. However, all data used in this manuscript can be made available upon request to the authors.

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
