# Peer review of "Coefficients of Thermal Expansion in La3Ga5SiO14 and Ca3TaGa3Si2O14 Crystals"

_materials, 2023, doi:10.3390/ma16124470_

Round 1

Reviewer 1 Report

In this paper, the authors have reported an experimental method on thermal expansion coefficients of La3Ga5SiO14 and Ca3TaGa3Si2O14 crystals, which are prepared by the Czochralski method. Utilizing the lattice change induced by temperature, the corresponding thermal expansion coefficients can be derived from XRD results. This work is interested and can be accepted after major revision based on the comments below.

1.      In Abstract, the temperature range is wrong. “25÷1000 °C” should be changed to “25-1000°C”. Similarly, “25÷800 °C”.

2.      “Previously, α was measured in the crystal Ca3TaGa3Si2O14 based on a study of the elongation of bars cut from the crystal as the temperature changes [26].” This sentence should be merged into the previous paragraph.

3.      “Thermal expansion coefficients ? must be taken into account when designing high-temperature sensors, acoustoelectronic and acousto-optic devices.” This sentence is wrong.

4.      The authors claim that “but only at temperatures above 800°C was there a significant change in the crystal surface, the surface became porous.” Only one SEM image of the crystals after heat treatment at 1000 °C was shown. Actually, more information on surface morphologies must be provided, especially below and above 800 °C.

5.      In Figure 5 and Figure 7 (only a few points), the experimental data is too small to calculate the thermal expansion coefficient. More experimental data are needed.

6.      here are many small peaks except most of the main perovskite structure, which maybe contribute the pyrochlore phase or other materials during the solid state reaction process. The authors should explain them clearly in text.

7.      In reference section, the formation issues can be found. For example, “Roshchupkin, D.; Ortega, L., Plotitcyna, O.; Erko, A.;…” in Ref 6. The authors must double check the references and correct them.

8.      There are some formation problems in text, as well as many grammar issues. The authors must carefully look through the manuscript and made a major revision.

1. There are some formation problems in text, as well as many grammar issues. The authors must carefully look through the manuscript and made a major revision.

Author Response

Dear reviewer, thank you for your helpful comments

  1. In Abstract, the temperature range is wrong. “25÷1000 °C” should be changed to “25-1000°C”. Similarly, “25÷800 °C”.

25÷1000 °C” and “25÷800 °C” were replaced by “25~1000 °C” and “25~800 °C”

  1. “Previously, α was measured in the crystal Ca3TaGa3Si2O14based on a study of the elongation of bars cut from the crystal as the temperature changes [26].” This sentence should be merged into the previous paragraph.

This sentence was merged into the previous paragraph.

  1. “Thermal expansion coefficients ?must be taken into account when designing high-temperature sensors, acoustoelectronic and acousto-optic devices.” This sentence is wrong.

«Thermal expansion coefficients  of the crystals of langasite family must be taken into account in the process of the fabrication of high-temperature sensors, acoustoelectronic and acousto-optic devices». It is very important for bonding.

  1. The authors claim that “but only at temperatures above 800°C was there a significant change in the crystal surface, the surface became porous.” Only one SEM image of the crystals after heat treatment at 1000 °C was shown. Actually, more information on surface morphologies must be provided, especially below and above 800 °C.

Figure 4. SEM image of LGS crystal surface before (a) and after heat treatment at 850°C within 30 minuts (b).

  1. In Figure 5 and Figure 7 (only a few points), the experimental data is too small to calculate the thermal expansion coefficient. More experimental data are needed.

Indeed, it is wrong to draw a linear dependence on two points. Three or four points already allow us to demonstrate the linear nature of the dependence. Additionally, we have demonstrated the possibility of measuring the coefficient of thermal expansion on the basis of measuring the rocking curves of an LGS crystal at different temperatures. This method is more precise.

  1. here are many small peaks except most of the main perovskite structure, which maybe contribute the pyrochlore phase or other materials during the solid state reaction process. The authors should explain them clearly in text.

The presence of small peaks is associated with the complex structure of crystals of the Langassite family, the presence of growth bands on which impurities are precipitated from the melt. Here it is possible both the presence of some phases in small quantities, and the presence of quasi-resolved reflexes. Here there is a significant difference in the intensity of even and odd reflexes. In general, there are a lot of interesting features, but so far everyone tries to avoid these questions. But it is an interesting task, where a combination of the whole set of X-ray diffraction and spectral methods and transmission electron microscopy methods is needed. In the future we will try to give answers to these questions.

  1. In reference section, the formation issues can be found. For example, “Roshchupkin, D.; Ortega, L., Plotitcyna, O.; Erko, A.;…” in Ref 6. The authors must double check the references and correct them.

Corrections have been made.

  1. There are some formation problems in text, as well as many grammar issues. The authors must carefully look through the manuscript and made a major revision.

We made a major revision of the manuscript.

Reviewer 2 Report

Roshchupkin et al reported the thermal expansion coefficients in La3Ga5SiO14 and Ca3TaGa3Si2O14 crystals prepared by the Czochralski method. The independent coefficients of thermal expansion of crystals ?? and ?? were determined by X-ray powder diffraction based on the analysis of X-ray diffraction spectra measured in the temperature range of 25~1000 °C. The results showed that in the temperature range of 25~800 °C the thermal expansion coefficients are linear. At temperatures above 800°C there is a nonlinear character of the thermal expansion coefficients associated with a decrease in the Ga content in the crystal lattice of the lantangallium silicate family crystals. This work is interesting and meaningful. I recommend publication of this work in Materials after some minor revisions:

1. In the abstract, “25÷1000 °C” and “25÷800 °C” should be replaced by “25~1000 °C” and “25~800 °C”, the authors should also check this error in the other parts of the manuscript.

2. English language and style are fine, but minor spell check will be required.

3. Please compare the results present in this manuscript with the reported analogues.

4. Most of the references are too old, so they should be updated by using very recent literatures.

Minor editing of English language should be required.

Author Response

Dear reviewer, thank you for your helpful comments

  1. In the abstract, “25÷1000 °C” and “25÷800 °C” should be replaced by “25~1000 °C” and “25~800 °C”, the authors should also check this error in the other parts of the manuscript.

25÷1000 °C” and “25÷800 °C” were replaced by “25~1000 °C” and “25~800 °C”

  1. English language and style are fine, but minor spell check will be required.

Colleagues from Germany corrected the scientific language. The title of the article was changed accordingly as Coefficients of thermal expansion in La3Ga5SiO14 and Ca3TaGa3Si2O14 crystals

  1. Please compare the results present in this manuscript with the reported analogues.

The values of the thermal expansion coefficients for the CTGS crystal calculated from expression (1) were  K-1 and  K-1, respectively. These values are in a good agreement with the values ( K-1 and  K-1) obtained in [26] when the change in the length of the CTGS crystal bar was measured at varied temperature conditions.

  1. Most of the references are too old, so they should be updated by using very recent literatures.

Unfortunately, questions about the use of crystals of the Langasite family are currently being solved in the commercial plane. Crystals are actively used, but there are constantly requests to measure physical properties of materials. Accordingly, earlier we accurately measured piezomoduli of crystals, in this article we measure coefficients of thermal expansion. We would like to show possibilities of application of X-ray methods for determination of physical quantities).

In the future there is a possibility to study acoustic properties of a crystal based on the analysis of X-ray diffraction spectra at different temperatures, which is important when creating high-temperature sensors, because with increasing temperature there is an increase in temperature.

Reviewer 3 Report

The  subject is interesting, however, the X-ray diffraction patterns at different temperatures must be refined by Rietveld method, in order to obtain data useful to compare the change of the unit cell parameters, then calculate the thermal expansion coefficients.

You must include at least one reference to compare your results, for example,  Kugaenko et al. Bulletin of the Russian Academy of Sciences. Physics, 2012, Vol. 76, No. 11, pp. 1258–1263. © Allerton Press, Inc., 2012.

You can find in the revised manuscript some general remarks. 

The comments can be found in the revised manuscript.

Author Response

(The authors gave the same response as above.)

Round 2

Reviewer 1 Report

After looking throughout this revised manuscript,  the authors hve addressed all my comments. Therefore, it can be accepted in currrent form.

Reviewer 3 Report

Thanks for attend the comments.